# Endometrial Cancer in Reproductive Age: Fertility-Sparing Approach and Reproductive Outcomes

**DOI:** 10.3390/cancers14215187

**Published:** 2022-10-22

**Authors:** Levent Mutlu, Diego D. Manavella, Giuseppe Gullo, Blair McNamara, Alessandro D. Santin, Pasquale Patrizio

**Affiliations:** 1Division of Gynecologic Oncology, Department of Obstetrics, Gynecology and Reproductive Sciences, School of Medicine, Yale University, New Haven, CT 06510, USA; 2IVF Unit AOOR Villa Sofia Cervello, 90146 Palermo, Italy; 3Division of Reproductive Endocrinology and Infertility, Department of Obstetrics, Gynecology and Reproductive Sciences, Miller School of Medicine, University of Miami, Miami, FL 33136, USA

**Keywords:** endometrial cancer, fertility-sparing, early stage, conservative management, progestin, hysteroscopy, assisted reproductive technology, oocyte cryopreservation

## Abstract

**Simple Summary:**

In this paper we addressed the challenges encountered by women diagnosed with endometrial cancers during their reproductive years. Depending on the genetic profile of the endometrial cancer, these young patients may benefit from fertility-sparing strategies, including hormonal, surgical and assisted reproductive technologies.

**Abstract:**

Endometrial cancer is the most common gynecologic malignancy in developed countries and approximately 7% of the women with endometrial cancer are below the age of 45. Management of endometrial cancer in young women who desire to maintain fertility presents a unique set of challenges since the standard surgical treatment based on hysterectomy and salpingo-oophorectomy is often not compatible with the patient’s goals. A fertility-preserving approach can be considered in selected patients with early stage and low-grade endometrial cancer. An increasing amount of data suggest that oncologic outcomes are not compromised if a conservative approach is utilized with close monitoring until childbearing is completed. If a fertility-preserving approach is not possible, assisted reproductive technologies can assist patients in achieving their fertility goals.

## 1. Introduction

Endometrial cancer (EC) is the most common gynecologic cancer in the United States with an estimated 66,570 new cases in 2021, and the cancer rates are increasing [1]. Although the median age at diagnosis is 63 years, 7% of EC are diagnosed in patients younger than age 45. The majority of patients with EC have disease localized to the uterus (67%) [2]. Analysis of the SEER database shows improved survival in younger patients, with low grade, and early stage disease [3]. The standard treatment of EC includes hysterectomy, bilateral salpingo-oophorectomy, and surgical staging. Therefore, treating EC in young patients presents a unique set of challenges, wherein competing interests of oncologic safety need to be balanced with the patients’ desire for future childbearing. In such circumstances, referral to a reproductive endocrinologist, shared decision-making involving the patient, their family, the oncological team, and counseling the patient on the risks of deviation from the standard surgical treatment, must be readily implemented. The limitations of current diagnostic methods such as endometrial sampling, imaging, and the overall limited amount of evidence on a fertility sparing approach should also be part of the discussion. From a medical care standpoint, a multidisciplinary approach involving gynecologic oncology, reproductive endocrinology and infertility, maternal fetal medicine, and genetics specialists will help to optimize the care provided to premenopausal patients with EC.

A large United States cancer database study evaluating outcomes in over 23,000 women with stage 1 EC suggests that treatment with fertility preservation using progestin therapy yields comparable 5-year survival outcomes to hysterectomy for stage IA disease, but not for higher stages [4]. Furthermore, despite existing reports of patients with stage 1A but higher-grade tumors (grade 2–3), there is not enough collective experience to support recommending fertility-sparing progestin therapy for these patients [5,6,7], limiting the use of fertility-preserving treatment to patients with stage IA, grade 1 EC only. These data highlight the importance of patient selection for fertility-sparing treatments.

In this review, we set out to characterize the evidence on diagnosis, treatment, surveillance, and fertility options for patients with early stage, low-grade EC. We discuss how to choose appropriate candidates for fertility-sparing treatment, diagnostic evaluation, surgical and hormonal treatment modalities, and surveillance. From the fertility approach, we discuss assisted reproductive technologies relevant to these patients.

## 2. Discussion

### 2.1. Candidates for Conservative Management

To assess the candidacy for the fertility-sparing approach, it is important to assess the endometrial cancer risk of extrauterine spread. In the landmark GOG-33 study, surgical and pathologic factors associated with cancer spread were studied in 621 patients with stage I or occult stage II endometrial carcinoma [8]. All patients in this study underwent a total abdominal hysterectomy, bilateral salpingo-oophorectomy, pelvic and periaortic lymph node dissection, and assessment of peritoneal cytology. The risk of pelvic LN metastasis increased with the grade and depth of invasion. Grade 1 tumors with inner third myometrial invasion had a 3% pelvic nodal metastatic risk, and the risk increased to 11% with deep myometrial invasion. On the other hand, grade 3 endometrial cancer had a 5% pelvic nodal metastasis risk even when confined to the inner third of the myometrium, which increased to 34% with a deep myometrial invasion [8]. A recent analysis of Surveillance, Epidemiology, End Results (SEER) data showed grade 1 EC without myometrial invasion had a 0.5% risk of pelvic nodal metastasis, compared with a 1.6% risk of nodal metastasis in grade 2–3 endometrial cancer without myometrial invasion [9]. Therefore, high-grade disease or endometrial cancer with myometrial invasion are considered contraindications for fertility-preserving approaches.

Fertility-sparing management in patients with EC is reserved for low-risk disease only. National Comprehensive Cancer Network (NCCN) selection criteria for conservative treatment in women wishing future childbearing include: (i) grade 1 endometrial adenocarcinoma with histology and grade confirmed on D&C, (ii) tumor confined to the endometrium by imaging studies (MRI preferred) (iii) no contraindications to hormonal therapy, and (iv) patients should undergo counseling that this is not the standard of care and the risks involved with pursuing these courses of treatment [10]. These recommendations are in line with the patient selection criteria delineated in the European Society of the Gynecological Oncology Task Force Recommendations for Fertility preservation [11]. British Gynaecological Cancer Society/Royal College of Obstetricians and Gyneacologists guidelines extend selection criteria to women with superficial myometrial invasion [12].

Most EC results from sporadic mutations; however, approximately 5–9% result from mutations associated with Lynch syndrome [13]. EC associated with Lynch syndrome occurs 10–20 years earlier than sporadic EC [14]. For patients with a strong family history of EC and/or colon cancer, germline genetic testing and referral to a genetic counselor is recommended, and this is especially important for women diagnosed with cancer younger than age 50 since Lynch syndrome is associated with an increased risk of secondary cancers such as colorectal or ovarian cancer [15,16]. Additionally, a proper genetic referral provides an opportunity for extending the testing to the relatives of the affected patient for risk counseling and surveillance of malignancies [10].

### 2.2. Diagnostic Evaluation

The classic presentation of EC is abnormal uterine bleeding or postmenopausal bleeding. Therefore, any patient with risk factors for EC should undergo diagnostic evaluation, which typically includes obtaining a clinical history, physical examination, imaging, and endometrial sampling. The prognostic factors of endometrial cancer, such as stage, grade, myometrial invasion, lymphovascular space invasion, and lymph node status are best evaluated after a hysterectomy. As this approach would not be compatible with fertility preservation, clinicians have to rely on the information obtained through endometrial sampling and imaging studies to estimate myometrial invasion, assess adnexal involvement, and risk of nodal involvement to use while counseling patients. Therefore, the limitations of these methods to accurately assess the disease status should be recognized and discussed with the patient.

Office endometrial biopsy has high sensitivity and specificity for diagnosing EC with a reported sensitivity, in premenopausal patients, of 91% as summarized in a recent meta-analysis [17]. However, office endometrial biopsy using a device such as a pipelle may have lower diagnostic accuracy compared with dilation and curettage (D&C). In a case series involving 482 patients with FIGO grade 1 EC on office biopsy, 17.4% of the patients that had been sampled with pipelle and 8.7% of the patients that had been sampled with D&C were upstaged on final pathology after staging surgery [18]. Notably, 1.7% of the patients with FIGO grade 1 on office biopsy were upgraded to grade 2–3/3 on final pathology, while no patient was upgraded to grade 3 after D&C. Although the sensitivities of these modalities are comparable for diagnosing FIGO grade 1 disease, the accuracy of D&C appears higher. This has implications in fertility-preserving counseling, whereby missing grade 3 disease would be associated with a higher risk of disease progression and is a contraindication to conservative management. The higher accuracy of D&C is likely related to the larger sample provided by this method. An additional benefit of D&C might be a more complete removal of disease confined to the endometrium. Hysteroscopic tumor resection and D&C have been suggested by several authors as a more targeted approach prior to initiating hormonal treatment [19].

Myometrial invasion is a prognostic factor for EC, and deep myometrial invasion increases the risk of nodal and parametrial involvement [20]. The imaging modalities such as transvaginal ultrasound and magnetic resonance imaging (MRI) are frequently utilized to assess the myometrial and cervical invasion and for ruling out an assessment of the adnexal involvement. Although an optimal imaging modality is not determined, MRI is usually considered the preferred modality by most experts [11]. The sensitivity of MRI at detecting myometrial invasion ranges between 70–95%, and specificity ranges between 80–95% [21,22,23,24,25]. A meta-analysis of imaging modalities demonstrated that contrast-enhanced MRI had increased diagnostic accuracy over non-contrast-enhanced MRI for detecting the myometrial invasion [26]. Data on transvaginal ultrasound compared to MRI has conflicting results and some studies report comparable sensitivity [27], whereas others reported higher accuracy with MRI compared to transvaginal ultrasound for assessment of the myometrial invasion [28].

### 2.3. Fertility Sparing Options for the Management of Endometrial Cancer

#### 2.3.1. Hormonal Treatment Modalities

The most studied hormonal treatment (HT) modalities for EC are progestin therapies such as medroxyprogesterone acetate (MPA), megestrol acetate (MA), and progesterone-releasing intrauterine device (IUD). Progestins have antiproliferative, stabilizing effects on the endometrium. The use of HT until completion of childbearing does not appear to be associated with worse oncologic outcomes in stage IA endometrioid endometrial cancer. In an analysis of the National Cancer Database from 2004–2014, women younger than 50 years old and with stage 1A EC treated with progestins had similar 5-year survival rates compared to women who underwent hysterectomy (97.5% 5-year survival in both methods) [4]. Contrary to excellent survival in stage IA patients regardless of the modality of the treatment, stage IB patients who were treated with progestins had lower 5-year survival compared to the surgical treatment (75% vs. 97.5%, hormonal treatment vs. surgery, respectively) [4]. Analysis of SEER data including 6339 women diagnosed with grade 1 or 2 and stage I endometrioid endometrial cancer between 1993–2012, showed 2.5% of the patients received HT and 97.5% underwent surgery. The cancer-specific death rate at 15 years was higher in the HT cohort than in the surgically treated cohort (9.2% vs. 2.0%, respectively). However, this difference was reduced substantially after propensity score matching and was not statistically significant when all causes plausibly related to EC were considered (11.9% vs. 8.69% HT vs. surgery cohorts, respectively) [29]. Patients’ specific risk factors or contraindications, such as breast cancer, myocardial infarction, pulmonary embolism, deep vein thrombosis, and smoking should be considered before systemic HT [10].

There is no consensus on optimal dosing or choice of agent, and the reported dose as well as the duration of treatment varies greatly in studies [19]. The dose of MA ranges from 10–400 mg, and MPA 20–1500 mg daily [30]. MPA 400–600 mg/day and MA 160–320 mg are the most commonly reported doses. Overall, the response rate to oral progestins ranges between 75–82% [30,31,32]. In general, complex atypical hyperplasia (CAH) has higher response rates, lower recurrence rates, and a lower risk of persistent disease compared to EC when treated with progestins. In a meta-analysis of 45 studies including 391 women with grade 1 stage I endometrioid EC the most commonly used HT were MPA (49%), MA (25%), levonorgestrel intrauterine device (19%) [31]. The median time to respond was 6 months, and CAH had higher response rates compared to EC (65.8% vs. 48.2%, *p* = 0.002). Similarly, a higher risk of persistent disease (25.4% vs. 14.4%) after HT, and a higher recurrence rate (35.4% vs. 23.2%, *p* = 0.03) were observed in women with EC compared to CAH.

In a prospective phase II study, 28 patients with grade 1, stage IA EC and 17 patients with atypical endometrial hyperplasia were treated with 600 mg MPA and low-dose aspirin for 26 weeks followed by continued cyclic estrogen–progesterone therapy until attempting pregnancy. Complete response and partial response rates in patients with EC were 55% and 32%, respectively, and three patients (14%) had no response. Among patients with atypical endometrial hyperplasia, 82% had a complete response, and 18% had a partial response to this regimen [33]. Among patients that responded to the treatment regimen, half of the responses were observed by 8 weeks and the remaining responses were observed by 16 weeks. During a median follow-up of nearly 48 months, 57% of patients with EC and 38% of patients with atypical hyperplasia had a recurrence. The recurrences were higher in patients with treatment-free periods and anovulatory cycles and were between 7 to 36 months. Body weight gain and liver function test (LFT) abnormalities were the most common toxicities, but no thromboembolic events were observed. Twelve pregnancies and seven deliveries were reported in the study population.

Levonorgestrel IUD (LNG IUD) systems release constant levels of progestins locally to the endometrium, with limited systemic absorption and minimal side effects; they have been increasingly utilized for the fertility-preserving treatment of EC. In a retrospective study of 48 patients receiving LNG IUD, complete response was observed in 89% of the patients with CAH (25 out of 28 patients), 81% of the women with grade 1 endometrial carcinoma (13 out of 16), and in 75% (3 out of 4 patients) with grade 2 endometrial carcinoma [34]. In another retrospective study of 46 patients, response rates to LNG IUD were 80% in patients with CAH, and 67% in patients with grade 1 endometrial adenocarcinoma. [35] A Cochrane review comparing LNG IUD releasing 20 micrograms daily to 10 mg MPA found insufficient evidence to support the superiority of either method in women with atypical endometrial hyperplasia [36]. In a recently published single-arm phase II trial, 21 patients with grade 1 endometrioid EC and 36 patients with CAH were treated with LNG IUD. Complete response was observed in 91% of the patients with CAH while disease progression was observed only in 6% during the 12-month follow-up. In patients with grade 1 endometrial cancer, 54% complete response, 13% partial response, and 20% progressive disease were observed at 12 months [37]. LNG IUD administered with a secondary HT has been investigated to assess whether the combination regimens could increase the response rates. In a prospective observational study, a combination of LNG IUD with MPA 500 mg/day showed an 87.5% complete response rate at 9 months [38]. In the follow-up phase II study, the same hormonal combination showed 37% complete response and 25% partial response for patients with grade 1 endometrioid EC at 6 months, possibly indicating the need for a longer duration of treatment [39]. GnRH agonist combined with LND IUD or letrozole showed an 88% response in patients with well-differentiated carcinoma in a pilot study; however, the utility of prolonged use of this therapy in a premenopausal population is limited given known risks to bone health [40]. Table 1 includes a summary of ongoing fertility-sparing clinical trials.

Given the association of insulin resistance and hyperinsulinemia with EC, metformin has been investigated as an additional therapeutic agent [41]. Metformin is shown to decrease gluconeogenesis, improve insulin resistance, decrease circulating insulin levels, and exert an inhibitory effect on the proliferation, and invasiveness of EC cells in preclinical studies [42,43,44,45,46]. Additionally, it may enhance sensitivity to progesterone treatment [45]. The addition of metformin to standard hormonal regimens has been investigated in several retrospective and prospective studies. In a phase II trial, 17 patients with CAH and 19 patients with grade 1 stage IA endometrioid adenocarcinoma were treated with MPA 400 mg/daily for 6 months and metformin 750 mg daily with gradual titration up to 2250 mg and continued during the study duration. Body mass index (BMI) was >25 kg/m^2^ in 25 patients and 67% had insulin resistance. During the 36-month follow-up, 81% of the study population had a complete response and the recurrence rate was 10% in patients with complete response, indicating that metformin can increase the response rates to MPA [47]. In a phase III trial, 150 patients with CAH or grade 1 stage IA EC were randomized to MA 160 mg/day or MA 160 mg/day plus metformin 500 mg three times a day. Overall, 40% of the patients were overweight and 40% had insulin resistance, measured by fasting glucose and insulin concentrations. At 16 weeks, a complete response rate was higher in women treated with MA plus metformin compared to MA alone (34% vs. 20%, respectively), although this result was not significant (*p* = 0.09). However, the effect of metformin was more pronounced among the CAH group, with complete response rates of 39.6% in the MA plus metformin cohort and 20.4% in the MA alone (*p* = 0.04) [48]. Trials evaluating the impact of metformin in combination with other HT are ongoing (Table 1, NCT02035787).

#### 2.3.2. Surgical Treatment Modalities

Hysteroscopic resection followed by an oral or intrauterine device (IUD)-releasing progestins as means to preserve fertility in patients with early stage EC has been widely recognized as an effective approach in patients wishing for future childbearing [49]. Indeed, existing data propose that women who received this treatment modality achieved the highest complete remission rate compared with other existing fertility-preserving treatment strategies [49,50,51,52,53]. Despite several published articles showing encouraging results for this combined technique, studies are limited to case series, case reports and retrospective data [52,54,55,56,57,58,59] summarized in a number of meta-analysis [32,50,60,61]. No randomized controlled trials have provided a sufficient level of evidence to make the combined conservative approach the standard of care [19]. Eligible patients wishing to preserve their future childbearing potential need to be aware that conservative treatment may be followed by hysterectomy with salpingo-oophorectomy to prevent disease relapse once family needs have been met.

The earliest reported case of fertility-sparing management of EEC was of a patient with Lynch syndrome who underwent hysteroscopic polypectomy displaying endometrial cancer after pathology analysis. Management included MPA for 3 months and the patient achieved a pregnancy 3 months after concluding her treatment [62].

The first published series using conservative surgery was by Mazzon et al. [57], describing six cases using a “three step” hysteroscopic resection technique consisting of resecting (i) the tumor lesion, (ii) the endometrium surrounding it, and (iii) the myometrium underlying the lesion, each step followed by pathology evaluation. Surgical resection was followed by 6 months of 160 mg/day megestrol acetate immediately after confirmation of a grade I EEC without myometrial invasion by pathology. The reported pregnancy rate was 65%, and all women achieved complete responses with no relapse after 50 months of follow-up. In another study of 14 women using the same hysteroscopic approach plus either MA or LNG IUD, Laurelli et al. [59], reported that 78% of patients reached complete response with no disease recurrence, achieving a 45% pregnancy rate without requiring assisted reproduction (ART). Giampaolino et al. [52], later suggested the addition of multiple random endometrial biopsies to Mazzon “three step technique” with levonorgestrel-releasing IUD inserted upon confirmation of EEC G1 with disease-free margins. The complete response rate in Giampaolino’s series was 78%, with a relapse rate of 2%, with no patients opting for pregnancy during the study follow-up period.

Alonso et al. [63], published a review of studies performed in reproductive-aged women with stage 1A grade 1 EEC treated with initial hysteroscopic resection followed by hormone therapy and found >85% response rate with 11% tumor recurrence, and an overall pregnancy rate of 22%, reaching 66% in those who underwent ART. A prospective series published by Falcone et al. [49], the largest to date, included 28 patients with stage 1A, grade 1, and 2 the underwent hysteroscopic resection + oral MA or levonorgestrel-releasing IUD. Complete response was observed in 25 patients (89%), with 2 showing persistent disease and 1 presenting with disease progression and requiring definitive surgery. This series found exceptionally high pregnancy and live birth rates reaching 93% and 86%, respectively [49]. More recent studies have further supported prior reports on the oncologic and reproductive outcomes of hysteroscopic resection + progestin therapy in EEC, with complete response rates of close to 90% in women with EEC (stage 1 A, grade 1), and pregnancy rates of 69% [61].

### 2.4. The Risk of Synchronous or Metastatic Ovarian Cancer

As previously discussed, the standard treatment of EC is surgical staging at the time of total hysterectomy and bilateral salpingo-oophorectomy. Ovarian preservation is an important consideration in the fertility preserving approach since the ovary is the source of oocytes for future pregnancy and oophorectomy is associated with numerous adverse effects such as the increased risk of cardiovascular disease, decreased bone mineral density, and troublesome menopausal symptoms [64]. On the other hand, the risk of synchronous ovarian cancer and metastatic endometrial cancer to the ovaries, as well as the hormonal effect of ovarian preservation on a neoplastic endometrium should be considered. According to NCCN Guidelines, ovarian preservation can be considered in select premenopausal women with EC with normal appearing ovaries, and no history of breast/ovarian cancer syndrome or Lynch syndrome [10]. Analysis of the SEER database patients aged <45 years with stage I EC did not reveal increased mortality rates with ovarian preservation [65]. Among more than 235,000 women with primary EC, synchronous ovarian cancer was seen in 1.7% [66]. In a case series, the majority of the women with synchronous ovarian cancer had grossly abnormal appearing ovaries [67]. The risk of ovarian micrometastases in low-grade apparent stage IA EC is low and reported to be around 0.4–0.8% [68]. There is no evidence that ovarian preservation is associated with an increased risk of recurrence in low-grade, early stage EC [69].

Among patients younger than 50 years diagnosed with EC, 5–9% have deleterious mutations associated with Lynch syndrome [70,71,72]. Compared to the general population, Lynch syndrome-associated EC occurs at a younger age (mean age 47–49 years). Patients with Lynch syndrome also have a younger mean age of diagnosis of ovarian cancer (42–49 years) and tend to have early stage disease diagnosis. Additionally, endometrioid and clear cell histologies are reported to be more common compared with the high-grade serous histologies among this population [70,73]. Synchronous EC has been reported in 22% of the women with Lynch syndrome-associated ovarian cancer [74].

### 2.5. Surveillance

Repeat endometrial sampling is recommended with a fertility-preserving approach to (a) assess the response to therapy and (b) assess for progression of disease or recurrence. There is no consensus as to the method of endometrial sampling or interval of endometrial samplings, however, close surveillance with endometrial sampling every 3–6 months with office pipelle biopsy or D&C are recommended by the NCCN [10]. Endometrial pipelle biopsy may be less reliable in comparison to D&C [11,75]. If a complete response is achieved, patients should be encouraged to not delay conception [10,11]. For patients not willing to conceive immediately, maintenance hormonal treatment with systemic progestin or LNG-IUD is advisable [11]. As previously discussed, according to current recommendations, patients should be counseled that the fertility preserving approach is not the standard of care for EC, and that hysterectomy, bilateral salpingo-oophorectomy with surgical staging is recommended after childbearing is complete or if progression occurs during surveillance. Additionally, if EC is still present after 6–12 months with progestin therapy, surgical management with staging is recommended [10]. In patients with failed hormonal therapy after 6 months, NCCN recommends pelvic MRI to exclude myometrial invasion and nodal metastasis [10].

### 2.6. Obesity and Weight Loss Management

Increasing rates of obesity among children and adolescents appear to be contributing to an increased incidence of EC among young patients [76]. Recently, investigators have evaluated if weight loss management could be a potential adjunct to HT among patients undergoing conservative EC treatment [77]. Attention on weight management is important for all overweight and obese patients undergoing treatment for EC, but appears especially crucial for those choosing conservative and uterus-sparing options, as unopposed estrogen from excess adipose tissue is a known risk factor for developing EC and HT treatment failure [78]. Authors have demonstrated that weight loss can decrease the recurrence of EC [79]. Investigators are starting to assess interest and utility in bariatric surgery as a tool for weight loss and its impact on HT for EC among young patients (Table 1, NCT04008563) [80]. Obesity is also an independent risk factor for infertility, [81] as well as a known risk factor in pregnancy [82]. Unique to our discussion, patients choosing fertility-sparing approaches to EC treatment would additionally benefit from weight loss approaches to improve chances of eventual conception and uncomplicated pregnancy.

Young patients with obesity and EC are also more likely to also experience poly cystic ovarian syndrome (PCOS) and diabetes [83]. PCOS is associated with infertility secondary to chronic anovulation as well as longer HT treatment duration for EC and shorter recurrence interval due to progesterone resistance [84,85]. While addressing obesity through weight loss management techniques can improve PCOS and infertility [86,87], providers should consider screening patients with obesity and EC for diabetes. Ensuring occult or newly diagnosed diabetes is treated properly can improve a patient’s chance of eventual conception and decrease pre-gestational diabetes related adverse pregnancy outcomes [88].

### 2.7. Molecular Characteristics and Potential Implications on Treatment

The genetic and molecular landscape of EC is becoming more and more important for both prognosis and treatment choice. Universal somatic testing is endorsed by the NCCN and is becoming standard-of-care [10]. This testing for mismatch repair stability proteins can be performed on pipelle biopsy specimens, D&C specimens, or final pathology specimens if staging surgery takes place. In a landmark paper by the Cancer Genome Atlas published in 2013, 373 endometrial carcinomas were analyzed using array and sequencing-based technologies, and 4 different clusters were identified. [89]. *POLE* ultramutated endometrial cancers carry an unusually high number of mutations with 232 × 10^−6^ mutations per megabase (mut/Mb); Microsatellite instability hypermutated (MSI-H) group with 18 × 10^−6^ mut/Mb, copy number low group with a group with lower mutation frequency (2.9 × 10^−6^ mut/MB) and most of the microsatellite stable endometrial cancers (MSS), and finally copy number high group with extensive somatic copy number alterations and low mutation rate (2.3 × 10^−6^ mut/Mb). *POLE* mutant tumors were found to have hotspot Pro286Arg and Val411Leu mutations in 76% of the samples in the exonuclease domain of DNA polymerase epsilon, which is responsible for DNA replication and repair [89]. Other significantly mutated genes in the *POLE* cluster were *PTEN*, *PIK3R1*, *PIK3CA*, *FBXW7*, and *KRAS* [89]. *POLE* mutant tumors have the best progression-free survival, whereas the copy number high tumors have the poorest outcome across the clusters. The second cluster is the microsatellite instability hypermutated group, which has a 10-fold greater mutation frequency than the microsatellite stable endometrioid tumors, with few somatic copy number alterations. Frameshift deletions in *RPL22* and non-synonymous *KRAS* mutations were observed, with few mutations in *FBXW7*, *CTNNB1*, *PPP2R1A*, and *TP53* [89]. MLH-1 promoter hypermethylation was observed in this cluster. Microsatellite instability can result from different mechanisms, including *MLH-1* promoter methylation, or germline mutations in mismatch repair genes such as *PMS2*, *MSH2*, *MSH6*, and *MLH1* as in Lynch syndrome or sporadic mutations in the mismatch repair genes (i.e., Lynch-like) [90]. Copy number low microsatellite stable endometrial cancer had unusually high *CTNNB1* mutation [89]. Additionally, increased progesterone receptors were found in this cluster, suggesting responsiveness to hormonal therapy. The final cluster is the copy number high cluster, which contains the majority of serous ECs, a quarter of grade 3 endometrioid tumors, with frequent TP53 mutations, and *FBXW7* and *PPP2R1A* mutations [89]. Importantly, cancer-specific survival was highest in the *POLE* ultramutated EC, followed by MSI-H, then copy number low EC patients. The copy number of high patients had the worst prognosis [89,91]. Notably, *POLE* mutant EC was mostly grade 3 on histologic examination, but some were grade 1 or 2, which underscores the importance of incorporation of the genetic/molecular landscape to histologic characteristics of the tumor to understand the prognosis [92]. There is an increasing amount of evidence in the literature supporting that recurrence is dependent on molecular characteristics, and the molecular background EC should be taken into account when making decisions with regard to adjuvant treatment to avoid potential overtreatment of some patients with favorable molecular subtypes (i.e., *POLE* ultramutated) [92]. The ongoing prospective randomized control trial PORTEC-4a is accounting for these molecular alterations in adjuvant treatment decisions [93]. At this time, there is not enough evidence to support extending molecular genotyping into fertility-preserving approaches, and further research is needed to address these questions.

Some studies have investigated whether molecular markers could predict response to HT. A systematic review of 27 studies with 1360 patients assessed the immunohistochemical markers that could potentially predict response to progestins [94]. Estrogen receptor (ER) and progesterone receptor (PR) were the most studied markers, and in some studies, high expression levels correlated with response to progestins [94,95,96]. Loss of PTEN expression alone did not predict poor response; however, when combined with low phosphor-AKT expression was associated with a poor response [97]. A single institution case series of 84 patients with CAH or FIGO grade 1 EC reported poor response to progestin treatment in patients with deficient MMR proteins (i.e., MLH1, MSH2, MSH6, and PMS2), but the deficient MMR cohort had only 6 patients, limiting the generalizability [98]. Overexpression of dual-specificity phosphatase (Dusp6) was associated with good response in patients with CAH compared with patients that did not have a Dusp6 overexpression [99]. Additionally, high expression of the endoplasmic reticulum stress glucose regulated protein 87 (GRP87) might be associated with poor response to progestin therapies [94,100]. Additional research will help to determine the best marker combination predicting response to hormonal therapies in patients with CAH and EC.

### 2.8. Fertility Preservation Options for Patients with Endometrial Cancer

Due to advances in treatments, the majority of EC cancer patients survive their diagnosis, with an overall 5-year survival rate greater than 95%. Consequently, fertility preservation and childbearing have become increasingly important for the quality of life after EC. Here are described the various options available to patients with EC through the use of assisted reproductive technologies [101].

#### 2.8.1. Assisted Reproductive Technology (ART)

The utilization and choice of ART for patients interested in fertility preservation depends on (a) the type and stage of cancer, (b) the treatment plan, (c) the time available until the cancer treatment has to start, and (d) whether the patient has a committed a partner. There are multiple, no longer experimental, options available to safeguard future parenthood. They include oocyte/embryo cryopreservation or ovarian tissue cryopreservation with the recent added potential benefit of in vitro maturation of immature oocytes recovered from the ovarian cortex [102]. In the event that complete staging surgery with a hysterectomy and bilateral salpingo-oophorectomy is considered the life-saving strategy and there is no time for a pregnancy, a patient can still preserve oocytes or embryos and then consider the use of a gestational carrier for achieving future parenthood [103].

#### 2.8.2. Oocyte Cryopreservation

Patients without a male partner or unwilling to cryopreserve embryos can preserve oocytes. Freezing oocytes, as opposed to embryos, avoid ethical and future legal concerns intrinsic to maintaining embryos in storage and perhaps the future need for disposal.

Improvements in the cryopreservation technique (vitrification rather than the slow-freeze protocol) have led to significant improvements in the overall outcome of oocyte cryopreservation, by reducing the cellular damage caused by ice crystal formation [104,105,106]. Recently, after rewarming, oocytes have reached survival and fertilization rates in excess of 75%, and live birth rates of 35% [107,108]. A number of ovarian stimulation protocols have been devised for fertility preservation. These protocols are individualized and specific for each patient’s unique diagnosis with the ultimate goal of optimizing mature oocyte yield. Progress in understanding the existence of multiple follicular maturation waves per menstrual cycle have allowed the implementation of ovarian stimulation protocols such as the “random-start protocol,” consisting in the administration of gonadotropin any day of the menstrual cycle, late follicular, peri-ovulatory or even in the luteal phase [109]. The main benefits of the “random-start protocol” are shorter time to complete the fertility preservation treatment (about 2 weeks instead of 4–5 weeks) and, furthermore, it can be employed for any patient, even for those who have intrauterine devices (IUD) in place. Leaving the IUDs in place has the added benefit of mitigating the estradiol-driven endometrial growth seen during ovarian stimulation. Another ovarian stimulation protocol is called “duo-stim” consisting of two egg retrieval procedures in a 28-day time frame, with the initiation of a second ovarian stimulation cycle 4 days after the first egg collection [110,111].

#### 2.8.3. Embryo Cryopreservation

For patients who have a committed male partner, embryo cryopreservation has a long track record of being a very successful procedure for fertility preservation [112]. Patients can expect excellent pregnancy rates from cryopreserved embryos, with success rates depending on a patient’s age at the time of embryo freezing. Even for embryo cryopreservation, mostly performed at the blastocyst stage, the most common methodology is vitrification. A meta-analysis published in 2008 confirmed the superiority of the embryo vitrification technique, therefore, replacing the slow freezing [113].

A typical treatment cycle for fertility preservation can be completed in 2 weeks from start to finish. The age of the patient and the number, stage, and quality of the frozen embryos mainly determine the future likelihood of successful live birth when performing embryo cryopreservation. Cryopreserved embryos have a very high (>95%) survival rate when rewarmed and the chances of a future live birth for women younger than 40 years old is about 40% [108].

For patients who are referred for fertility preservation during the luteal phase, gonadotropins can be started immediately (random-start protocol as described for oocyte cryopreservation) so as to keep the time to retrieval to no more than 2 weeks. Carrying out ovarian stimulation cycles during the luteal phase (as opposed to the follicular phase) has shown similar gonadotropin requirements, similar numbers of oocytes harvested, and similar fertilization rates [111]. An additional advantage of embryo cryopreservation for patients with EC as part of the Lynch syndrome spectrum is related to the possibility of offering preimplantation genetic testing to identify embryos carrying genetic mutations responsible for the syndrome.

#### 2.8.4. In Vitro Maturation

In vitro maturation (IVM) is a procedure adopted for maturing oocytes in the embryology laboratory when the oocytes collected are at the stage of germinal vesicles (GV’s) instead of metaphase II. IVM can be beneficial for patients with polycystic ovaries who are more at risk of developing EC (because of chronic anovulation) and at risk of ovarian hyperstimulation syndrome if exposed to standard ovarian stimulation regimens. Patients undergoing ovarian tissue freezing can also benefit from IVM if GV’s are identified during the ovarian cortex dissection. Immature oocytes are incubated in special culture media and if the oocytes mature in 24 h, as determined by extrusion of the first polar body, they can be cryopreserved or, if the patient had chosen to freeze embryos they can be inseminated, and the resulting embryos can be cryopreserved at the blastocyst stage.

This procedure of IVM is still inefficient when compared to standard ovulation induction protocols. Despite hundreds of live births, however, oocytes matured from GV’s produce fewer embryos with a lower chance of implantation, pregnancy, and live birth rate than conventional IVF [102].

#### 2.8.5. Gestational Surrogacy

As discussed in detail in this review, for some patients diagnosed with EC, a hysterectomy is strongly recommended. In these instances, it is still possible to use cryopreserved oocytes or embryos and establish parenthood with the help of a gestational surrogate. Gestational surrogates in general, are women that are compensated for their services or may act pro bono as when volunteering for a family member or a friend. In the last four decades, pregnancies with the use of gestational surrogacy, particularly for cases of absolute uterine factor infertility (such as post-hysterectomy) have been rising. It is also important to consider the geographical diversities among states in the USA. Some states are considered surrogacy-friendly granting pre-birth orders for the intended parents, while many other states are not friendly for the intended parents and require them to “adopt” their own genetic child. The friendly states are California, New York, Connecticut, Delaware, Maine, District of Columbia, New Hampshire, Nevada, Oregon, Rhode Island, and Washington. In the USA, gestational surrogacy is allowed and there are recommendations set forth by the American Society for Reproductive Medicine and rules on the testing set forth by the FDA. This is not the case for many other countries where gestational surrogacy is outright banned and as a consequence of these prohibitions, cross-border reproductive travel has become a known phenomenon [103,114]. In Sweden, for example, the banning of gestational surrogacy has led to the development of uterine transplants. Successful births have been reported by many teams, even in the USA (Texas, Ohio) who have been able to efficiently address not only the complexity of the surgeries, particularly when obtaining the uterus from living donors as opposed to deceased donors, but also the ethical/moral ramifications [115,116]. There are a number of requisites for people considering becoming gestational surrogates. They can be summarized as age (between 21–38); no medical contraindications to getting pregnant; BMI between 19–32; having already proven that they are capable to carry a pregnancy to term (no more than 5 spontaneous vaginal delivery or no more than three cesarean deliveries) without obstetrical complications; no smoking nor use of illicit recreational drugs; no history of child abuse, no convictions, and a clear background check [103].

In the US, gestational carriers are identified either through gestational surrogate agencies or directly by patients when family members or friends are volunteering for the service. A whole industry has developed that identifies and brings together potential gestational surrogates and intended parents. The internet has created a means for people to meet and pursue these arrangements with numerous national and international websites devoted to surrogacy [117]. It is also worth mentioning that finding a gestational carrier is the most time-consuming step of the treatment. At times, patients try to find a gestational surrogate through the internet and circumvent the use of agencies. This practice may expose intended parents to exploitation and disappointment. Being connected with a reputable surrogacy agency or a legal agent can alleviate many potential problems.

Whether through surrogate agencies or directly identified by intended parents, the gestational carrier has to be cleared medically and psychologically before being deemed eligible to carry a pregnancy for an intended parent. Medical clearance includes testing of the uterine cavity, a complete physical exam, maternal–fetal medicine consultation, and screening tests for sexually transmitted diseases (STDs). The preparation for the embryo transfer consists of using sequential incremental doses of estradiol tablets for about 14 days until the endometrium reaches a thickness of 7 mm or more as seen on ultrasound and the estradiol levels are 200 pg/mL or higher. At this time, the gestational surrogate is instructed to begin supplementation with progestin therapy six days ahead of the embryo transfer.

Gestational carriers should receive fair and reasonable compensation for their services in addition to health care coverage for the pregnancy and for any potential obstetric complications.

## 3. Conclusions

Fertility sparing approach for grade 1 endometrioid adenocarcinoma limited to the endometrium is feasible and can be considered in premenopausal patients with plans to conceive in the near future. The treatment should be individualized rather than a “one size fits all” approach and should be geared towards balancing oncologic safety with the individual patient’s goals. Patients should undergo extensive counseling about the risks, benefits, and alternatives to a conservative approach with understanding that the currently available data are largely retrospective case series with a wide range of treatment regimens and doses. Further prospective studies are needed to determine the optimal treatment regimens for premenopausal patients with EC.

## Figures and Tables

**Table 1 cancers-14-05187-t001:** Ongoing studies of fertility-sparing treatment of CAH and EC.

Study ID Number	Recruitment Status	Sponsor	Title	Study Type	Treatment	N	Disease	Duration
NCT02035787	Recruiting	UNC Lineberger Comprehensive Cancer Center	Metformin with the levonorgestrel-releasing intrauterine device for the treatment of complex atypical hyperplasia and endometrial cancer in non-surgical patients	Open label, single arm	Metformin 850 mg BID plus LR-IUD	30	CAH, grade 1 EC	12 months or until progression
NCT00788671	Active, not recruiting	M.D. Anderson Cancer Center	Levonorgestrel-releasing intrauterine system in treating patients with complex atypical hyperplasia or grade I endometrial cancer	phase II, single arm	LNG-IUD	69	CAH, Stage I–II grade 1 EC	12 months
NCT02397083	Recruiting	M.D. Anderson Cancer Center	Levonorgestrel-Releasing Intrauterine System with or Without Everolimus in Treating Patients with Atypical Hyperplasia or Stage IA Grade 1 Endometrial Cancer (LEVER)	phase II open label randomized trial	LNG-IUD vs. LNG-IUD + everolimus	270	CAH, Stage IA grade 1 EC	9 months
NCT04008563	Not yet recruiting	University Health Network, Toronto	Bariatric Surgery for Fertility-Sparing Treatment of Atypical Hyperplasia and Grade 1 Cancer of the Endometrium (B-FiERCE)	Randomized open label parallel assignment	Bariatric surgery + LNG-IUD vs. LNG-IUD	36	CAH, grade 1 EC	Evaluating recruitment rate only
NCT04792749	Recruiting	Samsung Medical Center	Clinical Effects of Metformin on Fertility-sparing Treatment for Early Endometrial Cancer	Non-randomized open label	Metformin 2000 mg + Medroxyprogesterone 500 mg + LNG-IUD	77	Stage IA grade 1 EC	12 months
NCT05316467	Recruiting	Xiaojun Chen	Weight Management Plus Megestrol Acetate in Early Stage Endometrioid Carcinoma	Open label, single arm	Megestrol Acetate 160 mg + lifestyle management	89	Grade 1 EC	2 years

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
