# Peer review of "Endometrial Cancer in Reproductive Age: Fertility-Sparing Approach and Reproductive Outcomes"

_cancers, 2022, doi:10.3390/cancers14215187_

Round 1

Reviewer 1 Report

Overall this is a well-written review on fertility-sparing approaches and reproductive outcomes for patients with endometrial cancer.

Line 35: Delete the word “endometrial, it is redundant.

Line 42-44: there appears to be overlap regarding medical teams with lines 37-38. Please condense.  

Lines 48, 77, 85, consider replacing the word “women” with patient to be more sensitive to those individuals who do not self-identify as a woman.

After paragraph 1, there is inconsistent use of EC for endometrial cancer.

Line 88, you do not have to spell out percent.

Line 90: please clarify what type of genetic screening (somatic vs germline) you are recommending. The two referenced papers do not actually recommend universal germline testing for ALL endometrial cancer patients. Family history is important and referral should be made to a genetic counseling for individuals with a strong family history of endometrial and/or colorectal cancer. Consider elaborating on the value of universal somatic testing on the cancer, which is endorsed by the NCCN and well supported by institutional data. Somatic testing for MMR proteins may be performed on initial biopsy, D&C or final hysterectomy specimen. Briefly highlighting the involved MMR proteins and a sentence or two about MLH1 hypermethylation.

Line: It is not clear that missing a grade 3 would actually be associated with a delayed diagnosis; it may be missed altogether due to sampling error. Consider deleting “associated to a delayed diagnosis” and reword, “associated with a higher risk of disease progression”.

Line 124: Consider clarifying “..increasing the response rate to progestin therapy”.

Lines 136-139 seems to overlap with lines 130-131. Please condense.

Line 162: Please be consistent with abbreviation for hormonal therapy (HT).

Lines 174: Please comment, if available on progression rate from CAH to cancer on resampling.

Line 193, need to spell out CAH if using it for the first time.

Line 199: please replace  “in to support” with “to support”.

Line 209, please be consistent with labeling the phase of studies (eg, phase II vs phase 2).

Line 212, even though GnRH agonst is mentioned in a pilot study, worth mentioning the implications of duration of therapy with such agent in premenopausal patients due to risk of bone health.

Line 232, was this statistically significant?

Line 235: recommend not using a new abbreviation EEC. Consider early stage EC.

Paragraph 272-275: Reasonable to combine with the following sentence and leaving out “in summary”.  I suggest moving up these sentences to the beginning of this section 2.3.2. so that readers can contextualize that the interventions in this section have insufficient evidence.

Line 297, please clarify, apparent Stage IA is in reference to the endometrial cancer correct?

Line 366: approach should be pleural.

Line 401: “Live-saving” should be “life-saving”

Line 410: There is an extra end parenthesis.

The section 2.7.5: the word, “States” does not need to be capital.

Line 493: insert “they” after “proven that”

Line 516: consider progestin therapy instead of progesterone

Line 530: there is an extra “the”

Line 532: Consider deleting “the”

Consider adding a section on weight loss management given that unopposed estrogen from weight is a major risk factor for many patients.

Reviewer 2 Report

In this article, the authors discuss the management of endometrial cancer in young women who desire to maintain fertility.  

In this article, obesity, an important public health problem, is not discussed. The increasing rate of obesity in children and adolescents appears responsible for the increasing prevalence of endometrial cancer in young patients. In particular, in obese patients, fertility-sparing management is associated with a lower probability of pregnancy [Gonthier et al. 2014]. 

In addition, a more detailed discussion on polycystic ovarian syndrome (PCOS) is necessary. PCOS is associated with a significant risk of endometrial cancer. Key mechanisms in endometrial carcinogenesis include both chronic anovulation (as the authors mentioned) and metabolic syndrome. Metabolic syndrom with a triad of obesity, hyperinsulinemia, and diabetes is commonly observed in PCOS [Shafiee et al. 2014]. PCOS is associated with longer treatment duration and shorter recurrence interval in patients receiving fertility-preserving treatment [Wang et al. 2021]. 

The authors discuss "Fertility Preservation Options for Patients with Endometrial Cancer." However, the discussion on oocyte cryopreservation and embryo cryopreservation is not adequate for endometrial cancer patients. These treatments are indicated for patients with malignancy who need to receive gonadotoxic systemic chemotherapy, such as breast cancer, lymphoma, and leukemia. Fertility preservation is not indicated for patients with advanced endometrial cancer who need to receive systemic chemotherapy using cytotoxic drugs.

There is no table in this manuscript. Tables are necessary To help understand the discussion.

CT is not used for the assessment of myometrial invasion, as MRI is more useful. 

What is the aim of this review? The authors need to state it in the Introduction.

Minor

Line 69. Spell out EC when it is first mentioned.

Round 2

Reviewer 2 Report

I cannot find Table 1 (Line 246).

The authors discuss fertility preservation options in greater detail. However, the abstract does not reflect this.

As the title of this article is "Endometrial Cancer in Reproductive Age: Fertility-Sparing Approach and Reproductive Outcomes", table(s) summarizing fertility-sparing approaches and reproductive outcomes in young women with endometrial cancer might be valuable. Readers (oncologists) may want to know this information.

Author Response

The abstract has been modified as suggested. 

The table has been reformatted and submitted now, as suggested by the reviewer.